# Synthesis of dienes from pyrrolidines using skeletal modification

Haitao Qin[1,2,5], Ting Guo[1,5], Ken Lin [1], Guigen Li[3] & Hongjian Lu [1,4] ✉

Saturated N-heterocyclic pyrrolidines are common in natural products, medicinal compounds and agrochemicals. However, reconstruction of their skeletal structures creating new chemical space is a challenging task, and limited methods exist for this purpose. In this study, we report a skeletal modification strategy for conversion of polar cyclic pyrrolidines into nonpolar linear dienes through a N-atom removal and deconstruction process. This involves N-sulfonylazidonation followed by rearrangement of the resulting sulfamoyl azide intermediates. This can be an energetically unfavorable process, which involves the formation of active C–C π bonds, the consumption of inert C–N and C–C σ bonds and the destruction of stable five-membered rings, but we have used it here to produce versatile conjugated and nonconjugated dienes with links of varying lengths. We also studied the application of this method in late-stage skeletal modification of bioactive compounds, formal traceless $C(sp^2)$–H functionalization and formal N-atom deletion.

The pyrrolidine scaffold is one of the most commonly encountered saturated N-heterocycles in biologically active molecules and natural products (Fig. 1a)[1,2]. This leads to increasing interest in the development of methods for the modification of pyrrolidines to access new chemical space. Functionalization of C–H bonds, which can modify the periphery of a molecule, is one of the most important and efficient methods for exploitation of such N-containing compounds (Eq 1, Fig. 1b)[3–6]. Alternative skeletal modification reactions, which selectively break internal inert C–C and/or C–N bonds and alter the core structure of a molecule, are still under development[7–10]. To date, most reported reactions have focused on single C–N bond cleavage and functionalization, commonly through the ring opening of quaternary-ammonium intermediates (Eq 2, Fig. 1b)[11–13]. Other synthetic strategies to modify the pyrrolidine skeleton have seen limited development. Recently, Sarpong et al. introduced a silver-mediated deconstructive strategy for transforming saturated N-heterocycles, including pyrrolidines, into halogen-containing linear amines via homolytic ring-opening (Eq 3, Fig. 1b)[14,15]. Deletion of the N-atom, a synthetic process that removes the N-atom of secondary amines and couples the remaining fragments,

is also highly desirable and has emerged as an efficient synthetic pathway for modifying the skeletal of pyrrolidines (Eq. 4, Fig. 1b)[16–20]. Levin et al. developed a useful method that deletes nitrogen from secondary amines, including one example of a pyrrolidine derivative, using an electrophilic N-pivaloyloxy-N-alkoxyamide via an isodiazene (1,1-diazene) intermediate[21,22]. Our laboratory found that sulfamoyl azides derived from secondary alkylamines could undergo a Curtius-type rearrangement, leading to an N-atom deletion process[23,24]. More recently, Antonchick et al. reported an approach to access the same isodiazene intermediates from iodonitrenes generated in situ and developed a synthetic method to produce cyclobutanes by deletion of the N-atom of pyrrolidines[25]. While recent developments in synthetic methods have provided new opportunities for diversification of pyrrolidine moieties, these approaches are still limited and new methods are required to further expand their scope. One potential strategy involves the oxidation of N-aminopyrrolidines to generate various products, including olefins (upper equation, Fig. 1c)[26], which could offer a new approach to the skeletal modification of pyrrolidines. However, the conventional synthetic sequence of N-nitrosation,

[1]Institute of Chemistry and BioMedical Sciences, Jiangsu Key Laboratory of Advanced Organic Materials, School of Chemistry and Chemical Engineering, Nanjing University, Nanjing 210093, China. [2]Jiangsu Key Laboratory of Neuropsychiatric Diseases and Department of Medicinal Chemistry, College of Pharmaceutical Sciences, Soochow University, Suzhou, Jiangsu 215123, China. [3]Department of Chemistry and Biochemistry, Texas Tech University, Lubbock, TX 79409-1061, USA. [4]The Key Laboratory of Functional Molecular Solids, Ministry of Education, Anhui Normal University, Wuhu, Anhui 241000, China. [5]These authors contributed equally: Haitao Qin, Ting Guo. ✉e-mail: hongjianlu@nju.edu.cn

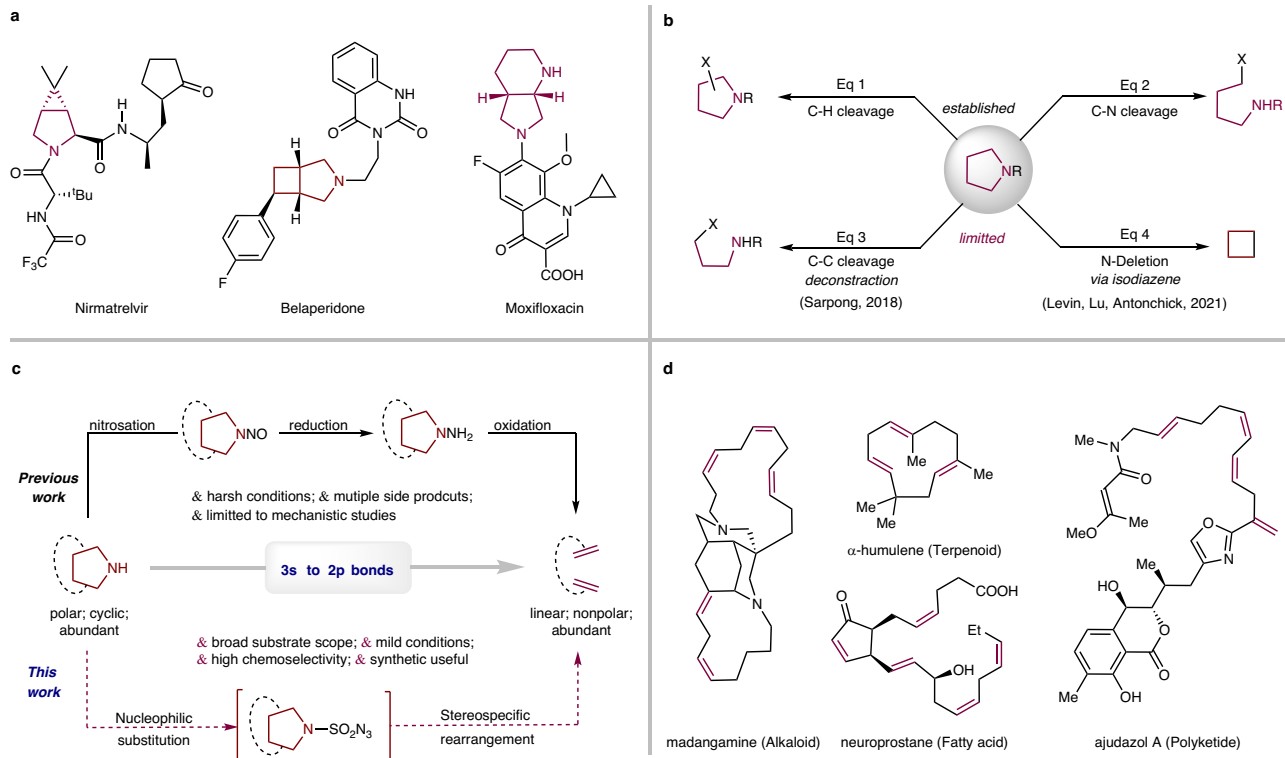

**Fig. 1 | Pyrrolidine and diene scaffolds. a** Selected drugs containing fused pyrrolidine fragment. **b** Pyrrolidine modifications. **c** N-removal and deconstruction of pyrrolidines. **d** Selected natural products containing conjugated or non-conjugated diene fragments.

reduction, and oxidation of the pyrrolidines is a multi-step and poorly chemoselective process that requires harsh reaction conditions and produces multiple products, often with olefins as minor products[27,28]. Consequently, this synthetic method limited to mechanistic studies and has failed to attract significant attention[26–28].

We report herein a method for conversion of pyrrolidines to dienes via a sequence of N-sulfonylazidonation followed by thermal rearrangement of the resulting sulfamoyl azide intermediates (lower equation, Fig. 1c). Currently, there are no general methods for the direct construction of two double bonds in conjugated and non-conjugated dienes and consequently, this approach is particularly attractive. The starting material, pyrrolidine and the resulting diene structures (Fig. 1a, d) are common in natural products and pharmaceuticals, and thus the reaction owns an abundant supply of substrates and versatile synthetic applications. However, this transformation presents a challenge, as it involves the formation of active C−C π bonds and the consumption of inert C−N and C−C σ bonds, as well as the destruction of a five-membered N-heterocycle.

## Results

The pyrrolidine-containing compound, azabicyclo[3.1.0]hexane, is a scaffold often used in drug discovery[29]. Accordingly, we chose azabicyclo[3.1.0]hexane (**1a**) as the substrate with which to test our hypothesis (Fig. 2a, see Supporting Information - SI for details). Our experiments show that **1a** reacts with a DCM solution of $N_3SO_2N_3$ to produce a sulfamoyl azide intermediate[30], which decomposes under thermal conditions to yield a 1,4-diene, ethyl (E)−2-styrylpent-4-enoate (**1b**) in an overall yield of 81% (entry 1). The use of a DCM solution of $N_3SO_2N_3$, generated from commercially available reagents $SO_2Cl_2$ and $NaN_3$, minimizes the associated purification and storage risks[31]. The reaction is highly stereoselective, and **1b** was obtained in an E configuration. None of the isomer (**1c**) with a Z configuration was detected. The excellent chemoselectivity of the reaction was demonstrated by the absence of the N-atom deletion product, cyclobutane (**1d**), whose

formation requires overcoming the strain in the four-membered ring. The yield of **1b** was reduced when the reaction was performed in the absence of ′BuOLi, at a lower temperature, or when dioxane was used as a solvent or the organic base DBU was used instead of ′BuOLi (entries 2–5). Imidazolium sulfonyl azide (Imi-$SO_2N_3$), developed by Fokin's group, has been used as an efficient sulfonyl azide transfer agent[32]. It can be prepared in crystalline solid form at a scale of tens of grams, with high yield and purity. Thus, two-step one-pot synthetic procedure has been developed by using 1.05 equivalents of imidazolium sulfonyl azide (Fig. 2b, standard conditions II - **SC-II**, see SI for details), affording desired product **1b** in 89% yield. This one-pot reaction, utilizing solid Imi-$SO_2N_3$ as the sulfonyl azide transfer agent, is easier to operate and more practical. To further study the stereoselectivity of the reaction, the isomers 2,3-*trans*-pyrrolidine (**2a**) and 2,3-*cis*-pyrrolidine (**3a**) were prepared and subjected to the standard reaction conditions (Fig. 2c). The reactions demonstrated exclusive stereoselectivity, yielding isomers **2b** and **3b** with E- and Z-olefinic structures, respectively. The stereo-chemistry of the products was determined by the structure of the starting materials.

The substrate scope of compounds with the azabicyclo[3.1.0] hexane scaffold was explored to assess the functional group tolerance and substituent group diversity (Fig. 3). The electron-donating and electron-withdrawing aryl groups at the α-position of the amino group were compatible with the reaction, which showed good yields for the production of 1,4-dienes (**4b-9b**) containing different substituted styrene fragments. Heteroaryl substrates, such as those substituted with furanyl (**10b**) or benzo-furanyl (**11b**) groups, were found to be particularly suitable. The reactions cleaved both benzylic C−N and alkyl C−N bonds synergistically. When the substrates had a *cis* stereochemical structure between the aryl group and the carbon chain link, the products (**4b-11b**) retained this *cis* stereochemistry, while substrates with a *trans* stereochemistry yielded the *trans* products (**12b**, **13b**). Substrates containing an alkynyl group at the α-position of the amino group were also good substrates, producing the desired

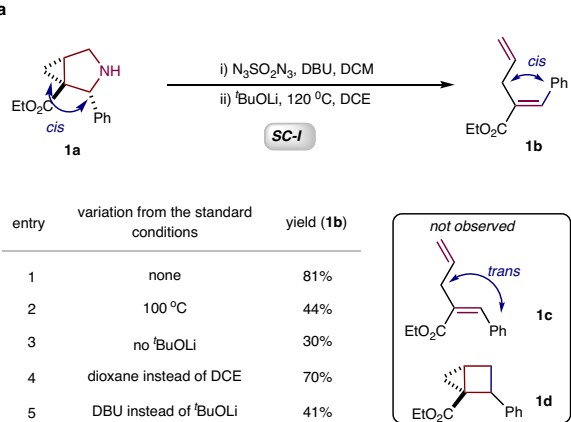

**Fig. 2 | Optimization of deconstruction of the pyrrolidines. a** Optimization of deconstruction of the pyrrolidine (**1a**) using N$_3$SO$_2$N$_3$. **b** One-pot synthetic procedure developed using imidazolium sulfonyl azide. **c** Complete stereoselective synthesis of diene isomers (**2b** *vs* **3b**). **SC-I**: (i) pyrrolidine (0.2 mmol), DBU (0.4 mmol), N$_3$SO$_2$N$_3$ (0.2 M in DCM, 3.0 mL), DCM (0.5 mL), rt, 3 h; (ii) $^t$BuOLi (1.0 equiv), DCE (2 mL), 120 °C, 3 h, isolated yield of diene based on pyrrolidine. **SC-II**: pyrrolidine (0.2 mmol), imidazolium sulfonyl azide (0.21 mmol), CH$_3$CN (2.0 mL), 0 °C, 3 h; Removing most of CH$_3$CN in vacuo, then $^t$BuOLi (0.2 mmol), DCE (2 mL), 120 °C, 3 h, isolated yield based on pyrrolidine.

products (**14b**, **15b**) by breaking the propargyl C–N and alkyl C–N bonds, respectively. Even relatively electron-deficient amines, such as an α-amino ester (**16a**) or an α-amino amide (**17a**), reacted under standard conditions to produce the desired 1,4-dienes in 72% (**16b**) and 82% (**17b**) yields. Substrates without substituents (**18a**, **19a**) or with a linear or branched alkyl group (**20a**, **21a**) at the α-position of the amino group also generated dienes (**18b-21b**) in good yields. Surprisingly, two strong alkyl C–N bonds were cleaved in the reaction[21–25]. A range of substituted olefins, including tri-substituted (**4b-11b**), 1,2-disubstituted (**12b**, **13b**), 2,2-disubstituted (**18b**), and mono-substituted (**19b**) olefins, were synthesized using this method. Polysubstituted dienes, such as the 1,2,4-trisubstituted-1,4-diene (**22b**) and the 1,2,3,4-tetra-substituted-1,4-diene (**23b**), were also produced. The method can generate various types of alkenes, including conjugated aryl alkenes (**4b-9b**), heteroaryl alkenes (**10b**, **11b**, **13b**), alkynyl alkenes (**14b**, **15b**), and electron-deficient acrylates (**16b**, **17b**). The link between the two double bonds in the products is varied, and can be a secondary carbon (**14b**, **15b**), a tertiary carbon (**23b**) or a quaternary carbon (**13b**, **16b**). The method can prepare dienes with links of varying lengths, including the conjugated 1,3-diene (**25b**) and the 1-4-diene (**26b**), the 1,5-diene (**27b**), and the1,6-diene (**28b**). A simple 1,7-diene (**29b**) was prepared in low yield, and a 1,6(7)-diene (**30b**) was produced from a bridge compound (**30a**) high efficiently. The method tolerates many functional groups, such as ester (**4b**), methoxyl (**5b**), fluoro (**7b**), chloro (**8b**), bromo (**9b**), heteroaryl (**10b**, **11b**, **19b**), amido (**17b**), alkynyl (**15b**, **24b**) and cyano (**19b**), providing various opportunities for the derivatization of the resulting dienes. Importantly, only a single alkene isomer was obtained in all cases. These synthetic transformations are significant as the resulting conjugated 1,3-diene and non-conjugated 1,4- (sometimes known as a skipped diene), 1,5-, 1,6-, 1,7-diene motifs are crucial intermediates in organic synthesis and ubiquitous in natural products[33,34]. Development of a direct and general method for the construction of two double bonds in one step is challenging, but this method achieves it without the use of oxidants or reductants.

## Synthetic applications

The synthetic applications of the N-removal and deconstruction process were studied, particularly for the late-stage diversification of complex molecules (Fig. 4a)[35]. Derivatives of Estrone and Stigmasterol were subjected to this method, and the desired 1,5-diene (**31b**) and 1,4-diene (**32b**) were produced in 92% and 74% yield, respectively. Bioactive compounds containing the pyrrolidine structure were also used in this transformation. Azabicyclo[3.1.0]hex-6-yl oxazolidinone (**33a**),

which has good antibacterial activity against gram-positive and gram-negative bacteria[36], was submitted to this reaction, and the desired diene derivative (**33b**) was achieved in 54% yield. Amitifadine (**34a**), an antidepressant drug, used directly as a substrate, provided the non-polar 1,4-diene product (**34b**) in adequate yield. A key fragment, **35a** of Belaperidone, containing an azabicyclo[3.2.0]heptane structure, was also identified as a suitable substrate, producing a 1,5-diene (**35b**) in a good yield. In combination with known methods, this synthetic methodology may also lead to the development of novel retrosynthetic routes. For example, Seidel et al. recently developed a direct α and/or β C–H functionalization of secondary cyclic amines[37–39] and this inspired us to develop a traceless C–H functionalization through a sequence of an NH-directed α-C–H functionalization and an N-removal and deconstruction process, providing various substituted dienes from commercial pyrrolidines (Fig. 4b). Seidel's reaction conditions were used for the C–H functionalization of bicyclic substrates without any further optimization. For example, the C–H (hetero)arylation of a commercial pyrrolidine (**36**) was followed by an N-removal and deconstruction reaction, providing 1,6-dienes containing aryl (**36b-38b**) or heteroaryl (**39b-40b**) olefin fragments in good to high yields. Utilizing C–H arylation of the commercial octahydroisoindole (**41**), a similar synthetic process enabled the synthesis of 1-phenyl-1,7-diene (**41b**). C–H alkylation of the commercial 3-azabicyclo[3.1.0]hexane (**42**) and a subsequent N-removal and deconstruction process could prepare a 1,4,12-triene (**42b**) with a satisfactory yield. The 2-alkynyl-5-methyl pyrrolidine (**43a**), formed from C–H alkynylation and subsequent C–H methylation of 6,6-dimethyl-3-azabicyclo[3.1.0]hexane (**43**), can be converted to a 1,5-disubstituted-1,4-diene (**43b**) under standard conditions. Through a sequence of C–H arylation, reduction of original olefin and N-removal and deconstruction, 1,3-*cis*-divinyl-cyclopentane (**44b**) could be obtained from the commercially available bridged compound (**44**). The cyclobutane fragment offers a rigid three-dimensional pharmacophore as a means of an "escape from flatland"[40]. Inspired by recently developed photochemical [2 + 2] reactions[41,42] and with N-atom deletion of pyrrolidines still in its infancy[20,25], a formal N-atom deletion was developed through a sequence of N-removal and deconstruction and photochemical [2 + 2] reactions (Fig. 4c). Combined with the method developed by Yoon[43,44], the formal N-deletion product bicyclo[3.2.0]heptane (**45c**), which has a cyclobutane fragment, was achieved in a total yield of 88%, providing a satisfactory and atypical pathway for the synthesis of fused cyclobutanes. This synthetic strategy can partially overcome a common issue in existing N-atom deletion methods for the synthesis of

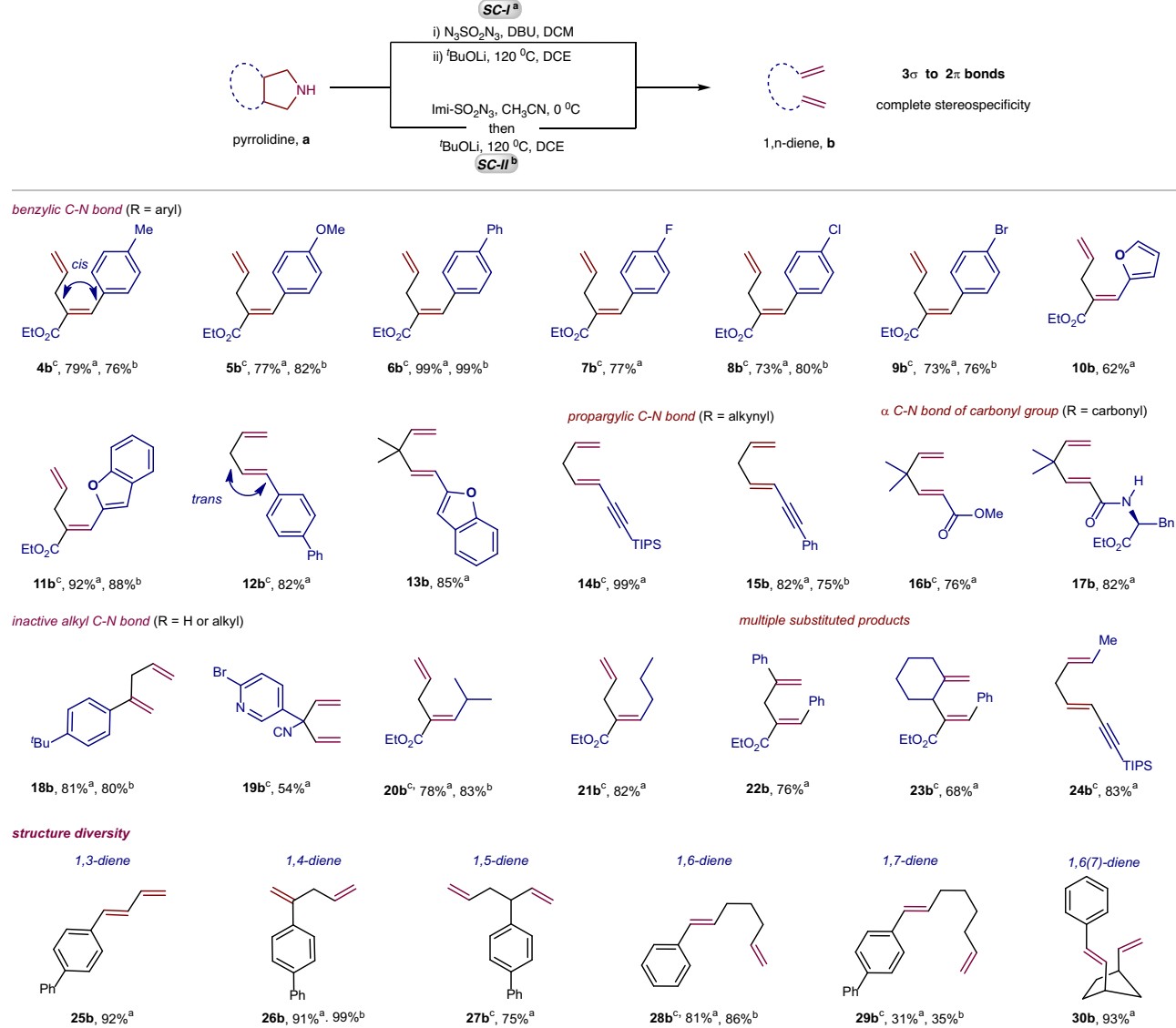

**Fig. 3 | Tolerance of functional groups, substituent group diversity and structure diversity in the N-removal and deconstruction of pyrrolidine scaffold.** [a]SC-I: (i) pyrrolidine (0.2 mmol), DBU (0.4 mmol), $N_3SO_2N_3$ (0.2 M in DCM, 3.0 mL), DCM (0.5 mL), rt, 3 h; (ii) [t]BuOLi (1.0 equiv), DCE (2 mL), 120 °C, 3 h, isolated yield of diene based on pyrrolidine. [b]SC-II: pyrrolidine (0.2 mmol), imidazolium sulfonyl azide (0.21 mmol), $CH_3CN$ (2.0 mL), 0 °C, 3 h; Removing most of $CH_3CN$ in vacuo, then [t]BuOLi (0.2 mmol), DCE (2 mL), 120 °C, 3 h, isolated yield based on pyrrolidine. [c] Using pyrrolidine·HCl instead of pyrrolidine, additional DBU (0.2 mmol) in SC-I or additional $Et_3N$ (0.2 mmol) in SC-II.

cyclobutanes[21–25], in which the substrate scope was limited and the olefins were often generated as side products. In addition, the gram scale of 1,4-diene (**45b**) was produced in 84% yield, demonstrating the practicality of this method.

## Mechanism studies

In order to investigate the reaction mechanism, several intermolecular radical trapping experiments were conducted (Fig. 5a). It was observed that the addition of three equivalents of radical trapping reagent TEMPO (2,2,6,6-tetramethyl-1-piperidinyloxy), BHT (2,6-di-tert-butyl-4-methylphenol), 1,1-diphenylethylene, or 9,10-dihydroanthracene had no significant effect on the reactions of **28a** or **29a**. The high-resolution mass spectrometry (HRMS) analysis indicated the potential existence of trapping products **28b'** and **29b'**, both of which contain two TMPO units. The reaction of 2-phenyl pyrrolidine **46b** under standard conditions was carried out with the use of diethyl fumarate to capture the intermediates. The HRMS analysis suggests that the formation of deamino cyclic product **46b'** is possible. N-Aminopyrrolidine (**45a'**)

was prepared to gain further insight into the reaction mechanism. The reaction of **45a'** under different oxidative conditions, which are known to oxidize 1,1-disubstituted hydrazines to the corresponding iso-diazene intermediates[45–47], failed to lead to the formation of the diene (**45b**) or cyclobutane (**45c**), judging by GCMS analysis (Fig. 5b). This demonstrates the advantages of our non-oxidative conditions. To further probe the mechanism, pyrrolidine (**47a**), bearing a cyclopropyl unit, was designed and synthesized. Its N-removal and deconstruction product (**47b**) was isolated in 89% yield, and no other detectable geometrical isomers were observed in the [1]H NMR spectrum of the crude reaction mixture (Fig. 5c, see details and crude [1]H NMR in SI). As depicted in Fig. 5d (see details and crude [1]H NMR spectrum in SI), the reaction chemoselectivity is associated with the $C_3$–$C_4$ bond in the pyrrolidine substrate. When this $C_3$–$C_4$ bond is part of a cyclopropane or cyclobutane ring, only N-removal and deconstruction products (**26b**, **27b**) were observed, suggesting that the released energy from the strained 3- or 4-membered ring opening processes may be responsible. Despite the decreased bent bond character of the $C_3$–$C_4$

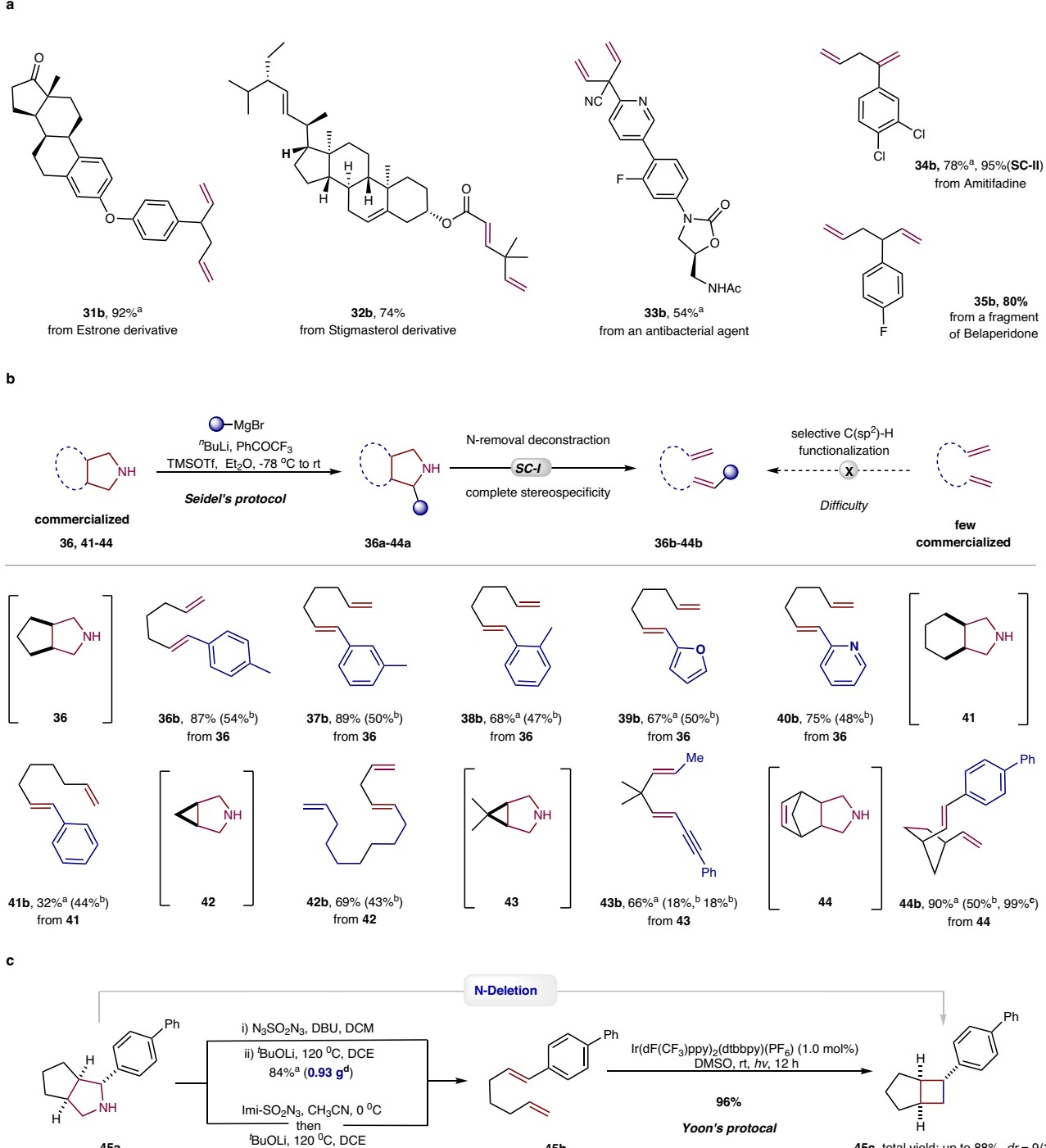

**Fig. 4 | Synthetic synergy. a** N-Removal and deconstruction of bioactive compounds. **b** A sequence of amine-directed C−H functionalization and N-removal and deconstruction reactions. **c** Formal N-deletion process of pyrrolidines. Standard conditions-I (**SC-I**): pyrrolidine **a** (0.2 mmol), DBU (0.4 mmol), N₃SO₂N₃ (0.2 M in DCM, 3.0 mL), DCM (0.5 mL), rt, 3 h; then *t*BuOLi (1.0 equiv), DCE (2 mL), 120 °C, 3 h, isolated yield of diene based on pyrrolidine. Seidel's protocol: *n*BuLi (2.0 equiv),

Et₂O, −78 °C, 10 min; PhCOCF₃ (1.2 equiv), −78 °C, 10 min; RMgBr (1.5 equiv), −78 °C; TMSOTf (1.2 equiv), −78 °C to rt, overnight. Yoon's protocol: Ir(dF(CF₃) ppy)₂(dtbbpy)(PF₆) (1.0 mol%), diene (0.2 mmol), DMSO, rt, *hν*, 12 h. [a]Using pyrrolidine•HCl instead of pyrrolidine, DBU (0.6 mmol). [b]Yield for the step of C−H functionalization. [c]Yield for the reduction step of the original double bond. [d]Gram scale preparation of the diene (**45b**).

bond in the cyclopentane ring, no notable decrease in stereospecificity or chemoselectivity was observed. A small amount of a cyclobutane product (**40c**) was observed, but N-removal and deconstruction products (**36b**, **40b**) predominated. Comparatively, the N-deletion reaction became the major reaction pathway when the C₃-C₄ bond was part of a cyclohexane fragment (**29b** vs **29c**, Fig. 5a). The reaction of 2-phenyl pyrrolidine **46a** led to the generation of both the

deconstruction product styrene **46b** and N-deletion cyclobutane **46c**, judging by GCMS and ¹H NMR analysis (Fig. 1d). These results suggest that the bent bond character of the reacting C₃-C₄ bond could significantly influence the chemoselectivity of reactions in which the N-removal and deconstruction process is dominant in most cases. Based on previous studies[21–25,48–50], a mechanism is proposed and is shown in Fig. 5e. Pyrrolidine reacts with N₃SO₂N₃ to form a sulfamoyl

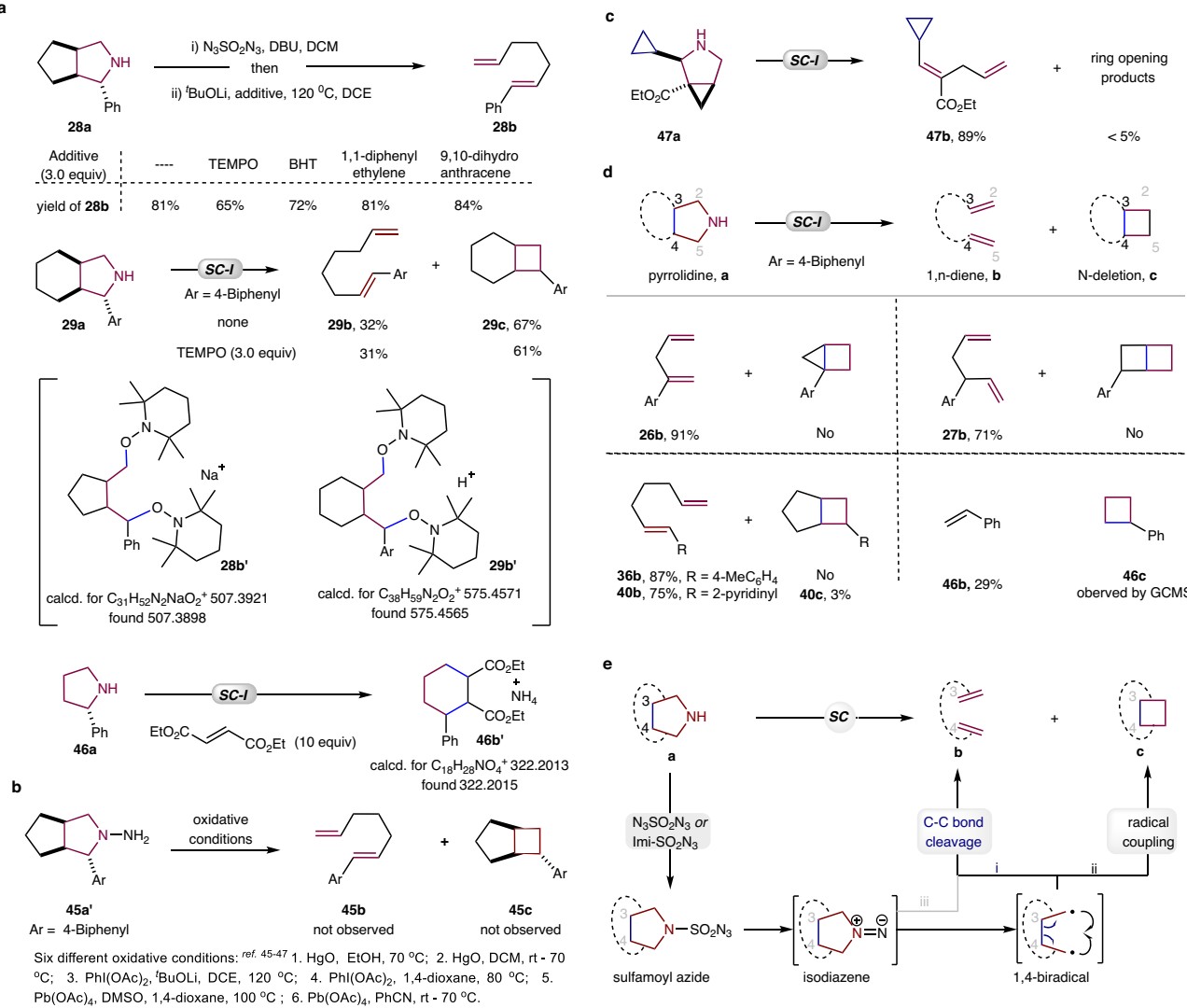

**Fig. 5 | Mechanistic Studies. a** Intermolecular radical trapping experiments. **b** The oxidation reactions of hydrazine (**45d**). **c** Cyclopropane as a radical probe. **d** Cyclobutane *vs* diene formation. **e** Proposed mechanism.

azide, which undergoes a Curtius-type rearrangement to generate an isodiazene intermediate. Its decomposition results in the formation primarily of a singlet 1,4-biradical intermediate. When the reacting $C_3-C_4$ bond is bent even slightly, rapid cleavage of $C_3-C_4$ bond occurs (path i), affording the dienes with excellent stereoselectivity and chemoselectivity. When the back bond is part of a cyclohexane, the N-atom deletion product was achieved via an intramolecular radical coupling reaction (path ii). However, a synergetic cycloreversion of isodiazene, a reaction in which three π bonds are formed and three σ bonds are consumed (path iii), can not be excluded[51,52].

We have presented a skeletal modification strategy for transforming cyclic polar pyrrolidines into linear nonpolar dienes, based on an N-sulfonylazidonation reaction and thermal rearrangement of the resulting sulfamoyl azide intermediate. This process does not require oxidants or reductants, enabling the synthesis of diverse conjugated and nonconjugated dienes with different substitutents and functional groups. Studies of the mechanism suggest that the process may involve a rapid $C_2-C_3$ σ bond cleavage from the thermally generated singlet 1,4-biradical intermediate as a key step, providing the dienes in excellent stereoselective and chemoselective fashion. The method provides a previously unknown pathway for the utilization of compounds containing pyrrolidine skeletons and for the synthesis of diverse conjugated and nonconjugated dienes, potentially leading to

new retrosynthetic routes and further advances in synthetic chemistry. In view of the large number of organic molecules containing diene and pyrrolidine fragments and the absence of general methods for direct construction of dienes, our synthetic strategy could be highly valuable. It provides a unique opportunity for the synthetic application of pyrrolidines and a general method for the direct synthesis of dienes with links of varying lengths.

## Methods
### General procedures I (SC-I)
To a solution of $N_3SO_2N_3$ (3 mL, 3.0 equiv, -0.2 mol/L in DCM) at 0 °C, the mixture of pyrrolidine **a** (0.2 mmol, 1.0 equiv) and DBU (0.4 mmol, 2.0 equiv) in the solvent of DCM (0.5 mL) was added dropwise. After the reaction was completed based on TLC (approximately 3 h). The solvent was removed under reduced pressure. The residue was then purified by quick flash column chromatography to obtain the crude sulfamoyl azide, which was subsequently added to a reaction vessel filled with ${}^tBuOLi$ (1.0 equiv, relative to sulfamoyl azide) and DCE (2 mL, 0.1 M) under argon atmosphere. After the mixture was stirred at 120 °C for 3 h with heating module, the reaction was cooled to room temperature. The solvent was removed under reduced pressure, and the crude mixture was purified by column chromatography to provide diene **b**.

## General procedures II (SC-II)

To a solution of imidazolium sulfonyl azide (0.21 mmol, 1.05 equiv) in the solvent of CH$_3$CN (2.0 mL) was added pyrrolidine **a** (0.2 mmol, 1.0 equiv) at 0 °C under argon atmosphere. After the reaction was completed based on TLC (approximately 3 h). Most of the solvent in the schleck tube is removed in vacuo. Then $^t$BuOLi (1.0 equiv, relative to pyrrolidine **a**) and DCE (2 mL, 0.1 M) was added into the schleck tube under argon atmosphere. After the mixture was stirred at 120 °C for 3 h with heating module, the reaction was cooled to room temperature. The solvent was removed under reduced pressure, and the crude mixture was purified by column chromatography to provide diene **b**.

## Data availability

The data supporting the findings of this study are available within the paper and its Supplementary Information files. Raw data are available from the corresponding author on request.

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

## Acknowledgements

Financial support for this work was provided by the National Natural Science Foundation of China (22071100 H.L., 22271148 H.L.).

## Author contributions

H.Q., T.G. and H.L. designed the experiments. H.Q. and T.G. performed the experiments and analyzed the data. G.L. participated in directing. All authors participated in writing the manuscript. H.L. conceived and supervised the project.

## Competing interests

The authors declare no competing interests.
