## [Peer Review File · Nature Communications]

REVIEWERS' COMMENTS

Reviewer #3 (Remarks to the Author):

The manuscript reported by Hongjian Lu and co-workers is centered on the development of a method for accessing alkenes with predictable stereochemical outcome from pyrrolidines. The reaction advances the chemistry previously developed by the same group (ref 23) that generates a sulfamoyl azide from a pyrrolidine and sulfonyl diazide. There are a few highlights in the methodology: i) In cases where the pyrrolidine core is readily available (e.g. in Figure 4b), the transformation has good synthetic utility for the synthesis of dienes; ii) the conversion of a pyrrolidine to diene opens the opportunity to access a cyclobutane (Figure 3c), thus illustrating the potential for skeletal editing via this chemistry.

This reviewer also noted some critical shortcomings:

- i) It is not clear why the title reaction would have a significant impact on skeletal editing. Pharmaceuticals containing a bicyclic pyrrolidine (e.g. compounds 33a-35a) require the pyrrolidine for biological activity and the diene obtained has very different properties (less hydrophilic, increased bond rotations etc.). Thus this pyrrolidine to diene transformation runs counter to the motivation behind skeletal editing for drug discovery, where the desired modifications to a molecular skeleton are often subtle (for a discussion, see *Science* 2019, 363, eaat0805). The authors provided only one example of how the diene can be utilized to construct a different molecular skeleton (Figure 4c).
- ii) The synthetic utility of the title reaction to synthesize dienes is limited because stereochemically defined pyrrolidines are still challenging to construct and require multi-step synthesis to access. Even though some pyrrolidines (e.g. in Figure 4b) can be made in one step from commercially available pyrrolidines, there is a limited number of the latter compounds. Furthermore, there are more synthetic methods to construct simple dienes than to functionalize pyrrolidines stereoselectively.
- iii) It was demonstrated in the literature (ref 26) that the conversion of isodiazenes into alkenes can go through both radical and non-radical pathways (i.e. cycloreversion). The authors' radical trapping and radical clock experiments were inconclusive because the results of the experiments could mean

either fast diene formation from a biradical intermediate or concerted cycloreversion. Thus no further clarification on the mechanism is provided in these studies. This reviewer also suggests changing the terms “N-atom deletion vs N-atom removal and deconstruction” (Figure 5d) to cyclobutane vs diene formation for greater clarity since the reader would generally interpret deletion and removal to have the same meaning.

Overall, even though there are synthetic advances reported in the manuscript, the reviewer is unable to recommend publication in Nature Communications.

Reviewer #4 (Remarks to the Author):

I have read the revised manuscript and the authors' responses to the reviewer 1's comments. The main concerns from this reviewer are the substrate scope, availability of the structurally specific pyrrolidines, and experimental safety. Two examples regarding 1,5-disubstituted pyrrolidines were provided; unfortunately, tetrasubstituted olefins cannot be shown. The authors trustily gave the reasons. I agree with them, and believe that the absence of this type of dienes doesn't affect the quality and importance of this interesting method.

In the case of the availability of the starting pyrrolidines, the authors showed the preparation of SM by C-H functionalization, which provides another useful way.

To address the safety problem, imidazolium sulfonyl azides were used, and one-pot method was explored, which would be good choice. In addition, I agree with the reviewer's concern regarding the use of DCM, which might be a problem.

For the question related to the use of BHT in the radical trapping experiment, the authors thought that tBuOLi reacted with ArOH to generate ArOLi that might function as a base. Nevertheless, 1 equiv of tBuOLi and 3 equiv of BHT were used, the pH value of the mixture would be acidic. How to explain the result?

Overall, based on the responses and some more examples were given, I think that most of the questions raised by the reviewer 1 have been properly addressed/explained.

Reviewer #3 (Remarks to the Author):

The manuscript reported by Hongjian Lu and co-workers is centered on the development of a method for accessing alkenes with predictable stereochemical outcome from pyrrolidines. The reaction advances the chemistry previously developed by the same group (ref 23) that generates a sulfamoyl azide from a pyrrolidine and sulfonyl diazide. There are a few highlights in the methodology: i) In cases where the pyrrolidine core is readily available (e.g. in Figure 4b), the transformation has good synthetic utility for the synthesis of dienes; ii) the conversion of a pyrrolidine to diene opens the opportunity to access a cyclobutane (Figure 3c), thus illustrating the potential for skeletal editing *via* this chemistry.

Response: *Thank you for summarizing some key highlights for our manuscript.*

This reviewer also noted some critical shortcomings:

i) It is not clear why the title reaction would have a significant impact on skeletal editing. Pharmaceuticals containing a bicyclic pyrrolidine (e.g. compounds 33a-35a) require the pyrrolidine for biological activity and the diene obtained has very different properties (less hydrophilic, increased bond rotations etc.). Thus this pyrrolidine to diene transformation runs counter to the motivation behind skeletal editing for drug discovery, where the desired modifications to a molecular skeleton are often subtle (for a discussion, see Science 2019, 363, eaat0805). The authors provided only one example of how the diene can be utilized to construct a different molecular skeleton (Figure 4c).

Response: *Thanks for the constructive suggestions. As the reviewer mentioned that skeletal editing for drug discovery means that the modifications to a molecular skeleton are often subtle. In this manuscript, the pyrrolidine skeleton has been converted to the diene skeleton, these two skeletons have very different biological activities. To avoid the misleading, we have made a modification to the "Skeletal editing" in the revised manuscript, renaming it as "Skeletal modification.". You can find the specific changes in the file labeled "manuscript with modifications".*

ii) The synthetic utility of the title reaction to synthesize dienes is limited because stereochemically defined pyrrolidines are still challenging to construct and require multi-step synthesis to access. Even though some pyrrolidines (e.g. in Figure 4b) can be made in one step from commercially available pyrrolidines, there is a limited number of the latter compounds.

Furthermore, there are more synthetic methods to construct simple dienes than to functionalize pyrrolidines stereoselectively.

Response: Thank you for your careful consideration. We acknowledge that the synthesis of stereochemically defined pyrrolidines can still pose challenges. However, many methods have been developed to synthesize complex pyrrolidines efficiently, such as [3+2] cycloaddition of imine with olefin (Hashimoto, T. & Maruoka, K. *Recent advances of catalytic asymmetric 1,3-dipolar cycloadditions.* *Chem. Rev.* **2015**, 115, 5366) and C–H functionalization of amines (Direct sp^3 C–H bond activation adjacent to nitrogen in heterocycles. *Chem. Soc. Rev.* **2007**, 36, 1069; C–H Bond Functionalization of Amines: A Graphical Overview of Diverse Methods. *SynOpen.* **2021**, 05, 173). These methods have been widely explored and have provided numerous opportunities for the synthesis of complex pyrrolidines. In this manuscript, we focused on one specific C–H functionalization method developed by Seidel's group (Direct α -C–H bond functionalization of unprotected cyclic amines. *Nat. Chem.* **2018**, 10, 165; α -Functionalization of Cyclic Secondary Amines: Lewis Acid Promoted Addition of Organometallics to Transient Imines. *J. Am. Chem. Soc.* **2019**, 141, 8778). This method allows for the direct α -C–H bond functionalization of unprotected amines, thereby simplifying the synthesis of complex pyrrolidines from commercially available pyrrolidines, as shown in Figure 4b. Converting a complexly synthesized pyrrolidine into a diene may not always be beneficial, especially if it involves multiple-step chemical transformations. It is important to carefully evaluate the potential advantages and disadvantages of such conversions in terms of synthetic efficiency, practicality, and yield. In scenarios where pyrrolidines are readily available or easily prepared (such as those derived from the [3+2] cycloaddition and C–H functionalization of pyrrolidines mentioned earlier), our method may indeed offer a useful strategy for transforming these fragments into dienes. As commented by reviewer #2 that “Lu and coworkers report a unique and clever approach to the synthesis of linear dienes from bi- or polycyclic N-H pyrrolidines, a nonobvious transformation that appears to be unprecedented.” we firmly believe that our translation of routine pyrrolidine synthesis protocols into a strategy for diene synthesis constitutes a significant and valuable contribution to the field of organic synthesis. Nonconjugated and conjugated dienes are not only common in natural products and pharmaceuticals, but they are also versatile intermediates in organic synthesis. Our method can produce dienes with various substituted groups and link lengths that are difficult to achieve through other known methods, making this transformation significant. In addition, the conversion of a pyrrolidine to diene opens the opportunity to access a cyclobutane (Figure 3c), thus illustrating the potential for skeletal editing via this chemistry. Comparing to the direct -deletion

method developed by Antonchick's group (*J. Am. Chem. Soc.* **2021**, *143*, 18864, in which the substrates were limited to the pyrrolidines contained aryl and carboxylic ester in the α -positions of nitrogen), the substrate used in our method does not have this limitation, thus provide a complementary pathway of skeleton editing pyrrolidines to cyclobutanes.

iii) It was demonstrated in the literature (ref 26) that the conversion of isodiazenes into alkenes can go through both radical and non-radical pathways (i.e. cycloreversion). The authors' radical trapping and radical clock experiments were inconclusive because the results of the experiments could mean either fast diene formation from a biradical intermediate or concerted cycloreversion. Thus no further clarification on the mechanism is provided in these studies.

Response: Thank you for carefully considering our work. In order to further investigate the reaction mechanism and also address the suggestions made by this reviewer, we studied the product distributions in the TEMPO capture reactions of cyclopentane- and cyclohexane-fused pyrrolidines (Fig. 5a), we find the radical capture products **28b'** and **29b'** contained two TEMPO units can be detected by HRMS. In addition, *e*-deficient olefin diethyl fumarate was used to capture the intermediate in the reaction of 2-phenyl pyrrolidine under standard conditions, the capture product cyclohexane (as a result of bi-radical intermediate coupling with olefin) can be detected by HRMS. These results are regarded as new evidence for the present of a biradical intermediate as we proposed in Fig. 5e.

This reviewer also suggests changing the terms "N-atom deletion vs N-atom removal and deconstruction" (Figure 5d) to cyclobutane vs diene formation for greater clarity since the reader would generally interpret deletion and removal to have the same meaning.

Response: As suggested, we have modified it accordingly in revised manuscript.

Overall, even though there are synthetic advances reported in the manuscript, the reviewer is unable to recommend publication in *Nature Communications*.

Response: We would like to express our sincere appreciation for your insightful comments, which have greatly contributed towards improving the overall quality of this paper. Based on the careful modifications made according to your and other reviewers' suggestions and taking into consideration the positive assessments from the other three reviewers, we hope the revised manuscript is now suitable for publication in *Nature Communications*.

Reviewer #4 (Remarks to the Author):

I have read the revised manuscript and the authors' responses to the reviewer 1's comments. The main concerns from this reviewer are the substrate scope, availability of the structurally specific pyrrolidines, and experimental safety. Two examples regarding 1,5-disubstituted pyrrolidines were provided; unfortunately, tetrasubstituted olefins cannot be shown. The authors trustily gave the reasons. I agree with them, and believe that the absence of this type of dienes doesn't affect the quality and importance of this interesting method.

Response: Thank you for the positive comments.

In the case of the availability of the starting pyrrolidines, the authors showed the preparation of SM by C-H functionalization, which provides another useful way. To address the safety problem, imidazolium sulfonyl azides were used, and one-pot method was explored, which would be good choice. In addition, I agree with the reviewer's concern regarding the use of DCM, which might be a problem.

Response: Thank you for your positive comments. We acknowledge the reviewer's concern regarding the use of DCM. In the newly developed conditions (SC-II, Fig. 2b), we have avoided the use of DCM.

For the question related to the use of BHT in the radical trapping experiment, the authors thought that tBuOLi reacted with ArOH to generate ArOLi that might function as a base. Nevertheless, 1 equiv of tBuOLi and 3 equiv of BHT were used, the pH value of the mixture would be acidic. How to explain the result?

Response: Thank you for your insightful comments. The pKa values of H₂SO₃ (resulting from the hydration of the side product SO₂) are approximately 1.9 and 7.0, and the pKa value of BHT (2,6-di-tert-butyl-4-methylphenol) is around 12.2. Therefore, it is possible that ArOLi could react with side product H₂SO₃, even in the presence of excess ArOH, leading to the formation of ArHSO₃ (and/or Ar₂SO₃) and ArOH.

Overall, based on the responses and some more examples were given, I think that most of the questions raised by the reviewer 1 have been properly addressed/explained.

Response: Thank you for your insightful comments and your kind decision.

REVIEWERS' COMMENTS

Reviewer #1 (Remarks to the Author):

The authors revised the manuscript in responses to the requests from all reviewers. Although DFT calculation was not added, other revisions were reasonably performed and additional data improved the quality of the manuscript. Now, I recommend publication of this manuscript as it stands.

Reviewer #2 (Remarks to the Author):

OK to publish, authors have fixed the issues.